# Validation of a Multi-Scale Contact Temperature Model for Dry Sliding Rough Surfaces

**Jamal Choudhry \*, Andreas Almqvist**  **and Roland Larsson**

Division of Machine Elements, Luleå University of Technology, 97187 Luleå, Sweden;
andreas.almqvist@ltu.se (A.A.); roland.larsson@ltu.se (R.L.)
* Correspondence: jamal.choudhry@ltu.se

**Abstract:** A multi-scale flash temperature model is validated against existing experimental work. The model shows promising results and proves itself to be a reliable tool for the accurate prediction of the flash temperature development between rough surfaces in sliding systems. Model predictions for the maximum flash temperatures as well as the bulk temperature fields were in very good agreement with the experimentally measured values. The model was also able to accurately predict the formation of hotspots as well as the temperature variations around the hotspots. From the model predictions, it is concluded that it is sufficient to only assess the flash temperatures on a small portion of the contact area and thus save both computational time and memory.

**Keywords:** multi-scale; validation; finite element method; flash temperature



## 1. Introduction

The concept of flash temperature rise in sliding systems is one of the most important topics in tribology. It refers to the instantaneous temperature rise that occurs due to two rubbing bodies in motion. Their origin can be traced to the small irregularities on surfaces that are called asperities. These highly localized hotspots play a significant role for the tribological behavior of rubbing surfaces because they can strongly influence the mechanical/chemical properties of the interacting bodies. It has been previously shown by Kennedy et al. [1] that the presence of localized hotspots can increase the risk for thermomechanical failure, cause thermoelastic instabilities and contribute to oxide formation. It has been shown by Quinn et al. [2] that flash temperatures can reach up to 1000 °C in asperities with contact lengths less than 10 μm. Several studies have been conducted to predict the flash temperature rise in contacts between rough sliding systems, such as the work of Blok [3] and Carslaw and Jaeger [4], who derived theoretical expressions to model the contact temperatures with moving heat sources. The work of Zeng et al. [5] and Komanduri and Hou [6] derived analytical expressions to model moving heat sources of different shapes. Another example is that of Bos and Moes [7] and Dinesh et al. [8] who derived expressions for the determination of the maximum temperature rise in simple circular and elliptical contacts.

To fully understand the development of the flash temperature rise in a sliding system, it is important to have a model that can account for each individual asperity contact. Kennedy et al. [9] presented analytical equations that use the concept of multiple heat sources to model individual asperities. In their analysis, however, they assumed that the asperities are sufficiently far apart so that no interaction happens between them, which limits their use in realistic contact scenarios. Most of the models that account for multiple heat sources are based on the work of Carslaw and Jaeger [4], which is based on moving heat sources in semi-infinite and insulated bodies. An example is that of Gao et al. [10], who developed a 3D transient flash temperature model that uses an FFT technique to speed up the numerical solution procedure, as well as that of Qiu and Cheng [11], who developed

a transient flash temperature model for mixed lubricated contacts. Another example is that of Waddad et al. [12], who presented a transient model for a rotating sliding system and also accounted for the presence of wear debris at the interface. The main drawback of using the method based on [4] is that it assumes that the bodies are semi-infinite in size as well as insulated, which might not be realistic in a typical sliding system. This is shown by Zhang et al. [13], where the accuracy of the Carslaw and Jaeger method is compared to the Fourier's law of heat conduction and it is shown that the solutions of Carslaw and Jaeger are only valid for very short sliding times. Another drawback with the previously mentioned methods is that they are not compared to any experimental work and are therefore not validated.

Fourier's law of heat conduction is an alternative approach to model the flash temperatures. The finite element method can be used to solve the Fourier's law of heat conduction in sliding systems as conducted by Ray et al. [14]. They developed a flash temperature model and compared their numerical results with experimental tests. Their analysis does not, however, give any adequate correlation between the experimental work and model predictions and also does not offer any clue of how the temperature develops in each individual asperities. The main advantage of using the finite element method is that it can be adapted to various types of contact regions as well as complex sliding systems, as shown by Choudhry et al. [15]. They presented a multi-scale flash temperature model that was validated against experimental work of a pin-on-disc machine and showed a promising method for determining the interface temperatures for complex sliding systems. Although they were able to correctly predict the macro-scale contact temperatures of a pin-on-disc machine, the predicted flash temperature results could not be validated due to the lack of experimental data. In general, there is a lack of validated flash temperature models. This is due to the models being either too unrealistic or use assumptions (such as boundary conditions and computation size domain) that are inappropriate for the problem. The lack of accurate experimental flash temperature measurements also makes it harder for models to be validated.

There have, however, been some attempts to experimentally determine the interface temperatures of sliding systems. Stephenson [16] and Kitagawa et al. [17] used thermocouples to determine the flash temperatures at the tool–chip interface during cutting. The presence of thermocouples can, however, disrupt the actual temperature of the tool and is therefore not considered an accurate method of measuring flash temperatures in sliding systems. Sutter and Ranc [18] conducted another kind of experiment to determine the flash temperature of a sliding system. Their methodology is based on the visible pyrometry technique developed by Ranc et al. [19]. The main advantage of their method is that it is non-intrusive and can accurately measure a wide range of temperatures.

The main objective of this paper is to validate the flash temperature model developed by Choudhry et al. [15] with the experimental results obtained by [18]. This was performed by recreating the same sliding system used in [18] and incorporating it in the model presented by the authors [15].

### 1.1. Research Questions

From the mentioned previous works, the following research questions arise:

- Is it possible to predict how the temperature develops and varies around the contacting asperities?
- Is it possible to only focus and assess the flash temperatures in a small portion of the contact region and still obtain accurate results?
- Is it possible to validate the numerical thermal model with experimental flash temperature measurements?

### 1.2. Originality and Significance of the Present Work

As mentioned previously, as of the writing of this paper, there is still no validated flash temperature model that can be used in sliding systems. The present work will demonstrate

the accuracy and validity of the model presented by [15] by comparing it with actual experimental results. The accurate prediction of flash temperatures as well as temperature fields in sliding systems can help to understand and reduce thermo-mechanical failures, prevent premature lubricant deterioration as well as optimize wear processes.

## 2. Methodology

An example of a multi-scale contact is shown in Figure 1. In the macro-scale contact, one body is stationary, while the other is moving with velocity *v*. Because of the increased complexity and size, the macro-scale contact will not incorporate the effect of surface roughness. The effect of surface roughness and asperity contacts will be considered only on a localized contact region called the "cell" and it is the micro-scale contact region. The cell's micro-scale contact region is thus the only region where the flash temperatures on the micro-scale are assessed. The main reason for only considering the effect of surface roughness at the cell's micro-scale contact region is that it allows for the macro-scale problem to be discretized with larger element sizes as compared to the micro-scale contact region, thus making it more computationally efficient.

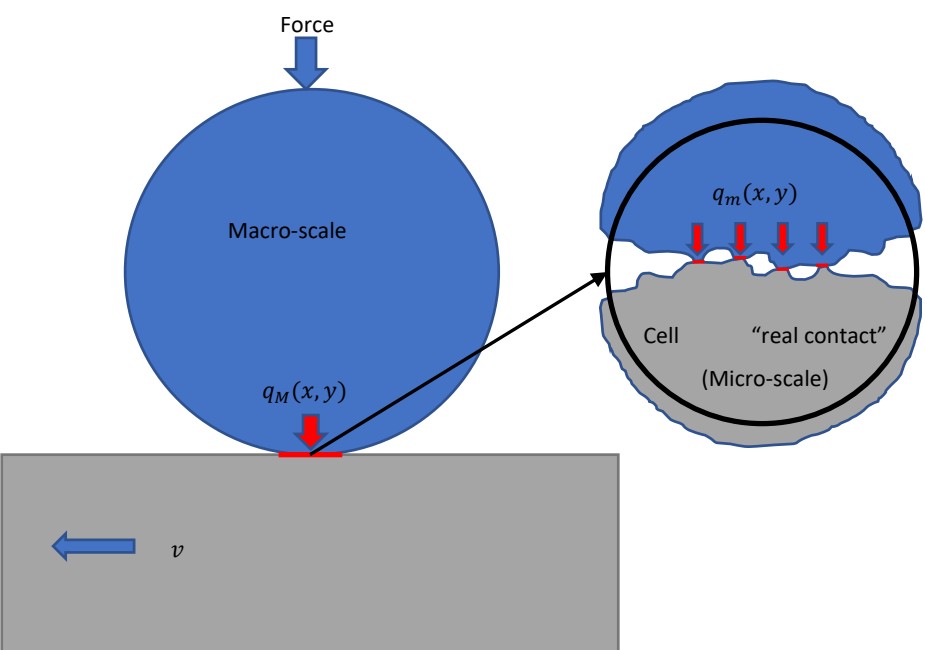

**Figure 1.** An example of the macro- and micro-scale contact regions. In the macro-scale contact, the upper body is stationary, while the lower body moves with velocity *v*. The red line indicates contact and $q_M$ is the spatially dependent thermal load generated due to the frictional force. The micro-scale contact region, i.e., cell, is the localized region where the effect of surface roughness is considered at the interface. The spatially dependent thermal load $q_m$ is generated due to frictional force from the asperity contacts.

The thermal load can be defined as the product of pressure distribution *p*, sliding velocity *v*, and friction coefficient *µ*, i.e.,

$$q = pv\mu. \tag{1}$$

In the case of a macro-scale contact between two rectangular bodies under a certain load, the macro-scale thermal load $q_M$ can be considered constant (due to the constant pressure distribution over the contact region). However, the micro-scale thermal load $q_m$ will always be spatially dependent, if the effect of surface roughness is considered there.

The full experimental details can be found in [18]. In short, an air gun was used to propel a projectile that was connected to the mobile specimen. The projectile was kept

nearly at a constant velocity during friction process and quasi-stationary conditions were achieved. The experimental setup adapted from [18] can be seen in Figure 2. The upper body (blue) is kept stationary, while the lower body (grey) has a prescribed movement. The upper body has a hole with a diameter 2 mm placed in the middle, which was used by the CCD camera to capture the flash temperatures [18]. The upper body has dimensions $h$ = 24 mm, $l$ = 12 mm, while the lower body has dimensions $w$ = 10 mm, $L$ = 60 mm. The purpose of the hole is to measure the temperature field of the lower body some time after it was in contact with the upper body. In other words, the hole will allow to display how the temperature on the interface in that particular area continuously develops as the lower body moves. Note that the hole is not a contact region, and its sole purpose is to reach the zone of friction and measure the flash temperatures.

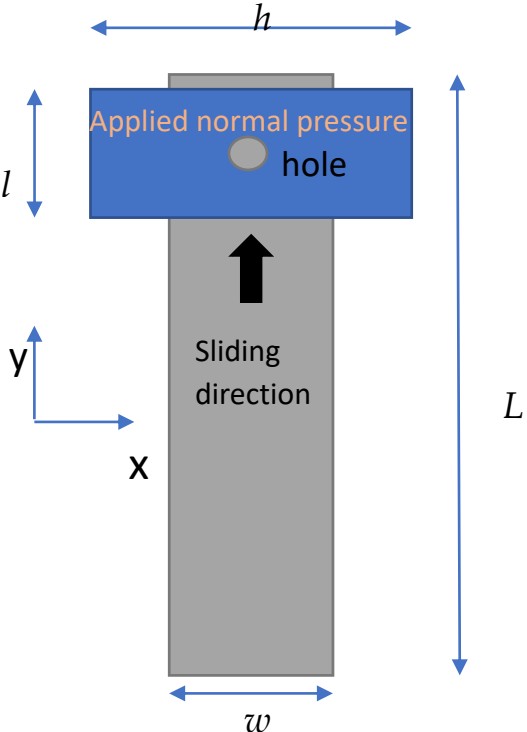

**Figure 2.** The experimental setup of the sliding system adapted from [18] (license No.:5264161172347). The upper body shown in blue is stationary, while the lower body shown in grey below is moving in the y direction.

The mesh and model geometry can be seen in Figure 3. The upper body has a square hole of size 2 mm × 2 mm placed right in the middle that reaches all the way down to the contact interface. The cell is placed right in front of the hole and the cell's interface (i.e., the micro-scale contact region where the surface roughness is considered and the flash temperature is assessed) has dimensions 2 mm × 1 mm. The micro-scale contact region has a much finer grid size as compared to the rest of the model; this is to accurately capture the temperature solution in the region of greatest interest. The micro-scale contact region within the cell is shown in Figure 4. A total of approximately 400,000 elements were used in the study and the micro-scale contact region was discretized into a 128 × 128 grid. The number of elements used in the model was determined in such a way that it does not affect the simulation results. The simulated time was set to 1000 μs, which was also enough time to ensure that steady-state conditions were reached. All thermal simulations were conducted in the Multiphysics software COMSOL® v.5.6 [20] with the operating conditions shown in Table 1. The same material properties were applied to both bodies and the properties are shown in Table 2. Because of the high temperature values reached within the contact, the thermal conductivity was selected to be temperature dependent during the analysis.

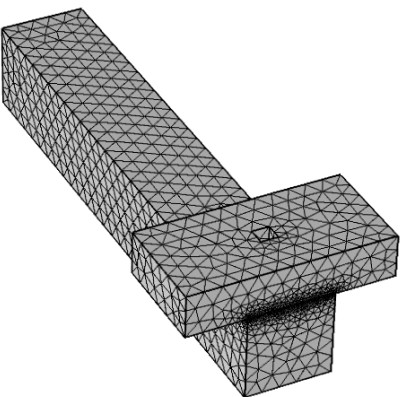

**Figure 3.** The mesh and model geometry of the sliding system.

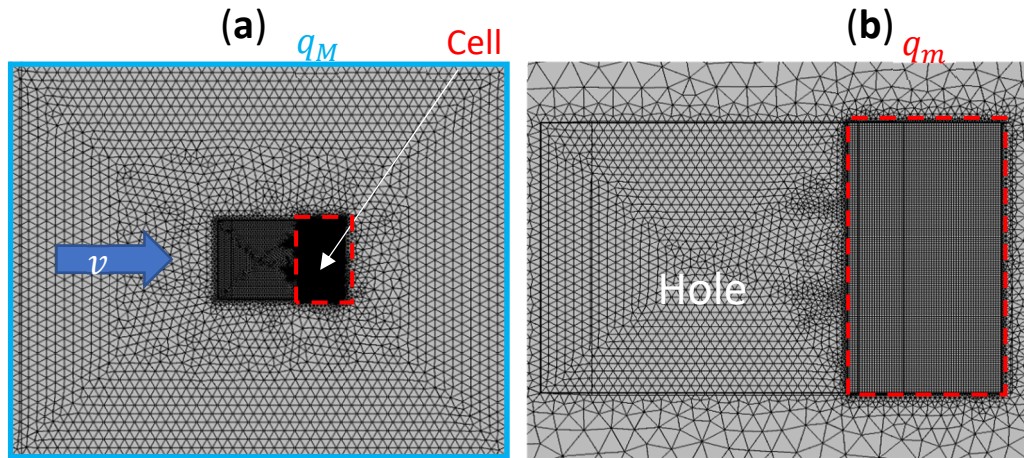

**Figure 4.** The macro-scale contact region seen from above and confined within the blue square as shown in (**a**) and the cell's micro-scale region confined within the dashed red box as shown in (**b**). The cell is placed directly to the right of the square-shaped hole (white dashed box) and the micro-scale contact region has dimensions 2 mm × 1 mm. The cell's micro-scale contact region is where the effect of surface roughness is considered and the flash temperatures are assessed.

**Table 1.** Operating conditions for the simulations based on the experimental work obtained from [18] (license No.:5264161172347).

| Simulation/Test | Sliding Velocity (m/s) | Normal Pressure (MPa) |
|---|---|---|
| A | 33.9 | 89.3 |
| B | 34.5 | 84.9 |

**Table 2.** Material properties of the bodies [18] (license No.:5264161172347).

| Temperature (degC) | Yield Stress (MPa) | Ultimate Tensile Strength (MPa) | Specific Heat Capacity (J/kg K) | Conductivity (W/m K) |
|---|---|---|---|---|
| 20 | 340 | 405 | 480 | 52 |
| 400 | 340 | 405 | 480 | 40 |
| 1000 | 340 | 405 | 480 | 30 |

The choice of the hole being square shaped instead of circular, as in the experiments, was made to accurately determine flash temperatures just behind the cell in a more convenient way. The shape of the cell's micro-scale contact region is rectangular, thus it allowed

for the hole to be placed directly behind the cell without any gap due to the circular shape. A heat flux boundary condition is prescribed to the micro-scale contact with the heat fluxes $q_m$ computed by the relation:

$$q_m(x,y) = \mu p_m(x,y)v \,, \tag{2}$$

where $\mu$ is the friction coefficient, $p_m$ is the pressure distribution and $v$ is the sliding velocity. In the contact region outside the cell (i.e., macro-scale contact region), a constant boundary heat source $q_M$ is applied according to the equation:

$$q_M = \mu p_M v, \tag{3}$$

where $p_M$ is the nominal normal pressure. Finally, the following relation holds between the micro- and macro-scale thermal loads:

$$\int_A q_m dx dy = q_M, \tag{4}$$

where $A$ is the macro-scale contact area. The spatially dependant pressure distribution $p_m$ in the micro-scale contact region is obtained by a method developed by Salhin, Almqvist et al. [21] and the contact simulation was performed in MATLAB. Due to the very short sliding distances, the effect of wear was not considered and all the boundaries in the model were thermally insulated. The nominal pressure $p_M$ was obtained from Table 1. The friction coefficient $\mu$ varied extensively in the experiments and was calculated by dividing the tangential force with the normal force of test 1 [18]. The friction coefficient is shown in Figure 5.

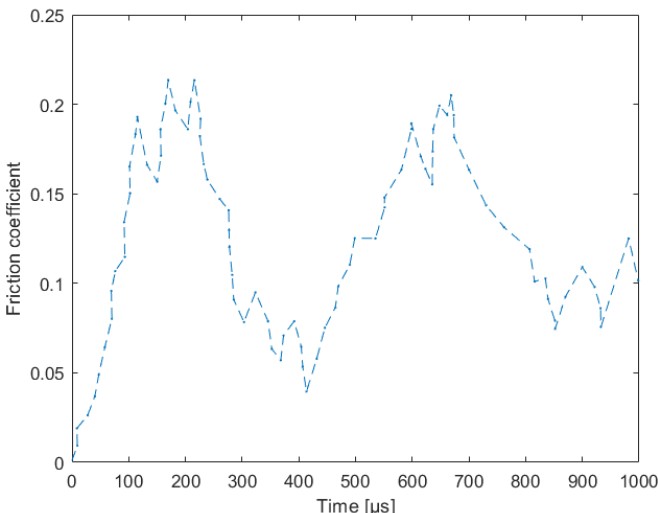

**Figure 5.** The friction coefficient used in the numerical model and obtained from [18] (license No.:5264161172347).

The surface topography of the bodies was generated by a method developed by Pérez-Ràfols and Almqvist [22], where random surfaces are generated based on a given height probability distribution and fractal characteristics of the frequency content. The surfaces used in the experiment were grinded in the sliding direction with an average roughness height of $R_a = 0.8$ μm [18]. Because the full details of the surface topographies used in [18] were not known, it was necessary to generate and simulate several surface topographies with different roughness parameters. The main reason for doing this was to match the predicted temperature fields as closely as possible with the ones measured in [18]. The main roughness parameters that were varied are: high frequency cut-off; anisotropy ratio; and finally, the Hurst exponent. The high frequency cut-off sets the limit for the shortest

allowed wavelength in the surface, while the anisotropy ratio is the ratio of the frequency distributions between the x- and y-directions. An example of the surface topographies used in the numerical model are shown in Figure 6. Note that, as in the experiments presented in [18], both bodies were given surface topographies with the same roughness parameters.

**Table 3.** An example of the surface roughness properties used in the numerical model.

| Assigned to Body | Surface Type | High Frequency Cut-Off | Anisotropy Ratio in the Sliding Direction | Hurst Exponent | Average Roughness Height $Ra$ [μm] |
|---|---|---|---|---|---|
| Upper | 1 | 0.01 | 1.5 | 0.6 | 0.8 |
| Lower | 2 | 0.01 | 1.5 | 0.6 | 0.8 |
| Upper | 3 | 0.01 | 2 | 0.8 | 0.8 |
| Lower | 4 | 0.01 | 2 | 0.8 | 0.8 |
| Upper | 5 | 0.02 | 2 | 0.8 | 0.8 |
| Lower | 6 | 0.02 | 2 | 0.8 | 0.8 |

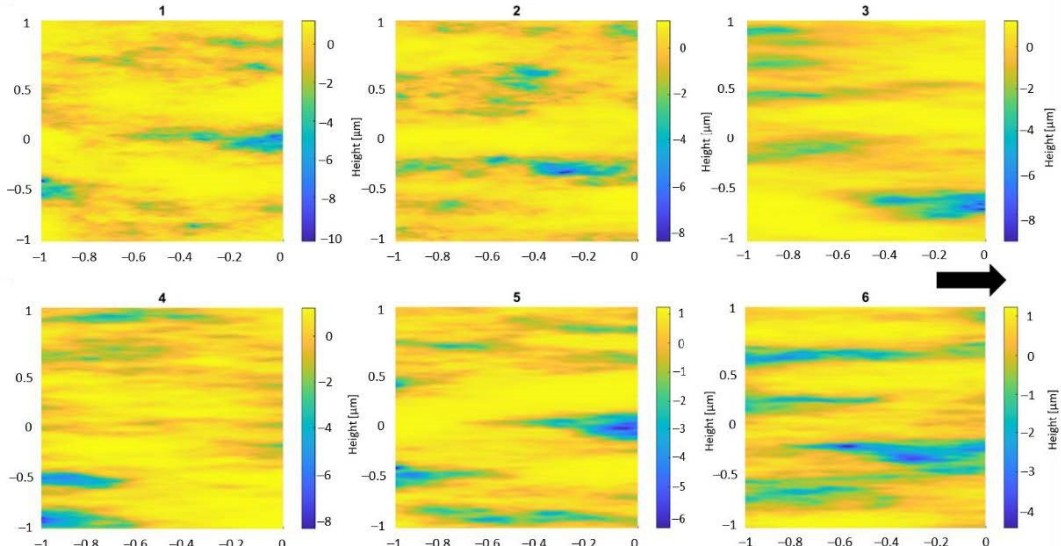

**Figure 6.** Example of surface topographies used in the numerical model. The size is 1 mm × 2 mm and the arrow indicates the sliding direction. The corresponding roughness parameters are described in Table 3.

The surface topographies were generated with a Weibull height probability distribution [20] and given an anisotropy ratio greater than 1 in the sliding direction, in order to resemble a surface that is grinded in the sliding direction. The reason for choosing a Weibull height probability distribution was that the surfaces in the experiments were grinded, and this height distribution is commonly used to model surfaces of this type [22]. The rest of the roughness parameters were chosen such that the predicted temperature field would match the measurements as closely as possible.

To briefly summarize, the general methodology is based on [15] and is as follows:

1. Define the macro-scale problem as shown in Figure 2.
2. Place the cell (region of interest for flash temperature assessment) just in front of the hole (Figure 4b).
3. Define the surface topography with the method presented in [22].

4.     Solve the contact mechanics problem over the micro-scale contact region with the method presented in [15].
5.     Use the contact pressure distribution from step 3 and apply it as input in the micro-scale contact region (Figure 4b) of the thermal model using Equations (2) and (3).
6.     Apply the thermal model and compare the numerical results with the experimental results.

### 3. Results and Discussion

A comparison of the temperature field of simulation A between the different surfaces is shown in Figure 7. As it can be seen, the temperature field can vary extensively depending on how the surface roughness is chosen. The result shown in Figure 7a is for surface type 1 and 2 (See Table 3), whose temperature fields were considered to best resemble the measurements (in terms of number of streaks, the "shape" of the streaks and maximum flash temperature). For this reason, the simulations with surfaces 1 and 2 will be from here on forth presented and compared with the measurements.

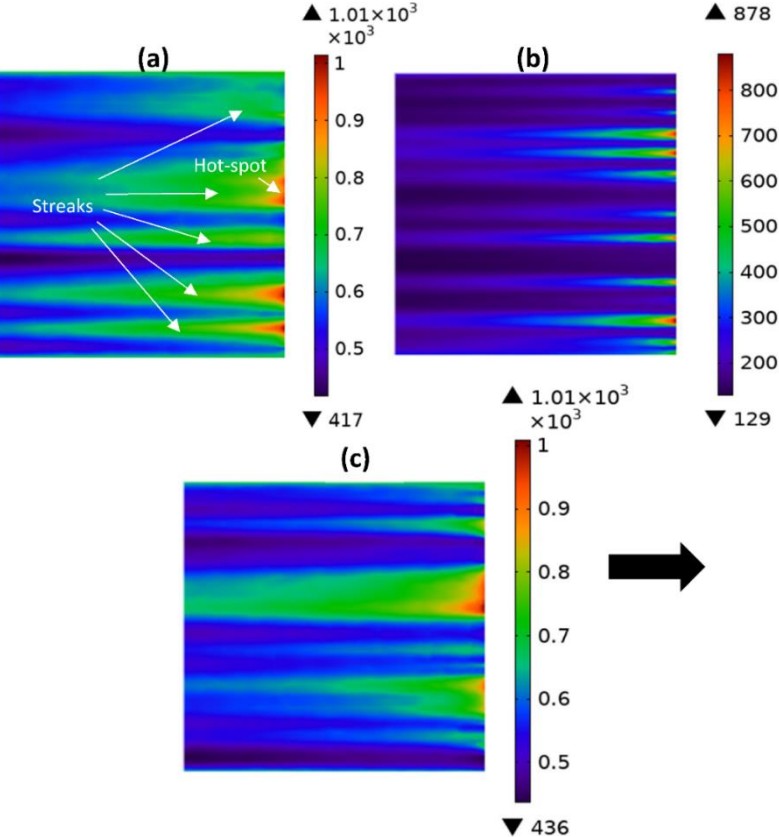

**Figure 7.** The temperature field of the hole of simulation 1 for three different example surface topographies. The temperature field is taken at time t = 650 μs and corresponds to the maximum average temperature obtained. In (**a**), the simulation is for surfaces 1 and 2; in (**b**), the simulation is for surfaces 3 and 4, and in (**c**), the simulation is for surfaces 5 and 6. The arrow indicates the sliding direction. The temperature has units in degree Celsius.

Figure 8 shows the comparison between the predicted temperature fields of simulations A and B. As it can be seen, the maximum temperature is higher for simulation A. This is in agreement with the measured temperature fields of [18] (See Figures 9a and 10a). The reason for this is that the nominal pressure is higher and thus the load for each individual asperity is increased. This behaviour is clearly illustrated in Figure 8 since, even though the same surface topographies were used for simulations A and B, the temperature field of

simulation A has a slightly larger heated contact area as compared to simulation B. Because there is no contact in the hole interface, no hotspots can be seen there. Indeed, the hotspots should only occur at the contact region where the asperities of the interacting bodies collide. A comparison between the predicted and measured temperature field for the holes of simulation A and test A is shown in Figure 9. As it can be seen, there is a good agreement between the two fields. The maximum of measured and predicted temperatures is 1004 °C and 1010 °C, respectively. It can also be seen that the hotspots leave behind a "tail", which is due to the high sliding velocity of the lower moving part. A comparison between the predicted and measured temperature field for the holes of simulation B and test B is shown in Figure 10. It can be seen that a good agreement between the two fields is observed here as well. The maximum of measured and predicted temperatures is 918 °C and 899 °C, respectively. A comparison between the hotspot temperature evolution along the *x*-axis is shown in Figure 11. Even here, a good agreement can be seen, indicating that the model can correctly predict the hotspots. Figure 11b shows the comparison between the shapes of the hotspot of streak 7. As it can be seen, the hotspot of simulation A reaches a higher temperature compared to simulation B, which is in agreement with the measurements. Finally, the temperature evolution along the sliding direction of a hotspot inside the hole can be seen in Figure 12. Although the predicted and measured temperatures are in good agreement near the hotspot, there are deviations introduced further away from the hotspot. The deviations here can be due to the fact that the exact surface topography is not known, so it would be impossible to exactly re-create the same contact conditions of the experiments. Depending on where the asperities lie and how big the individual contact spots are, the temperature evolution along the sliding direction can vary extensively. The present model predictions only rely on the average surface roughness of 0.8 µm given in [18], but other surface roughness parameters, such as the Hurst exponent and the high cut off frequencies, also play a significant role on how the temperature distribution evolves, as shown in [15]. It should also be noted that the temperature decrease along the sliding direction should follow an exponential decay as predicted by the numerical model. The temperature decay measured in [18] seems to behave linearly, which might be unreasonable. This erratic behaviour may be related to the thermal interference from the stationary body, which may also emit thermal radiation waves and thus introduce errors in the measurements. Nonetheless, a good overall agreement was observed between the model predictions and measured values, which presents the flash temperature model as a reliable tool.

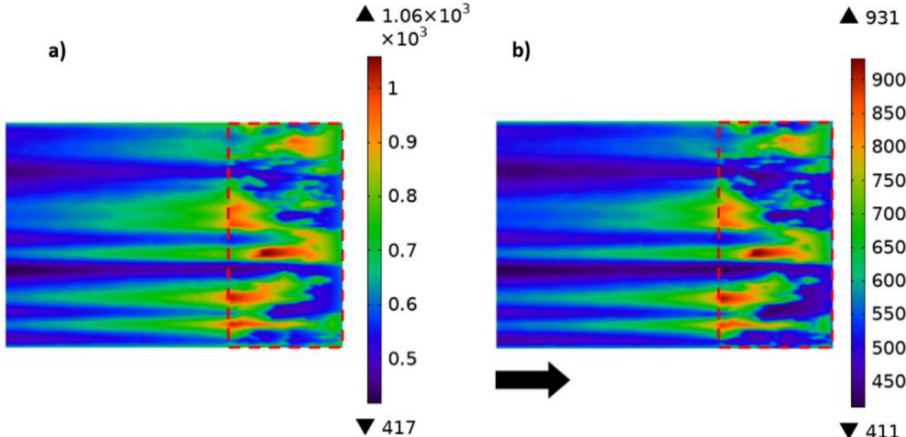

**Figure 8.** The predicted temperature field of the hole as well as micro-scale contact region (confined within the red dashed box) for (**a**) simulation A and (**b**) simulation B. The surfaces used are 1 and 2 and the simulated time is t = 650 µs. The temperature has units in degree Celsius.

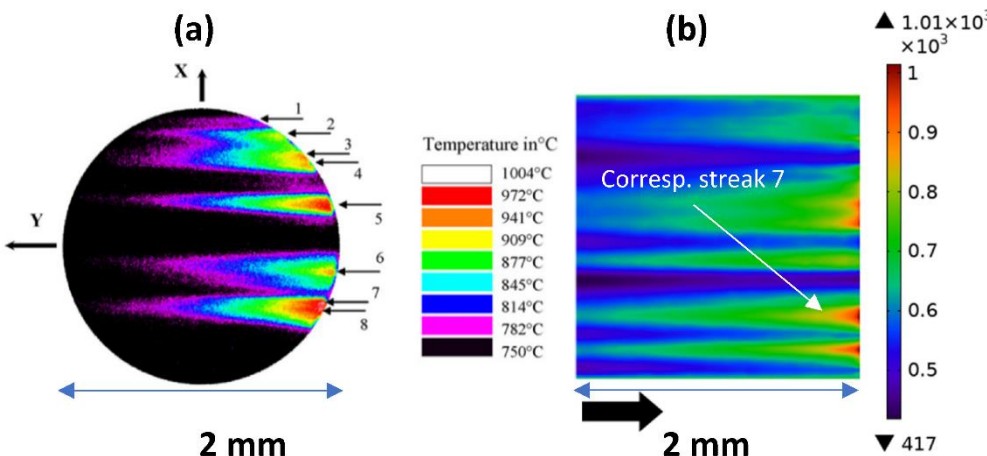

**Figure 9.** Comparison between the measured temperature field for the hole of test A obtained by [18] (license No.:5264161172347) shown in (**a**) and the hole of simulation A shown in (**b**). The simulated and test time is 650 μs. The temperature has units in degree Celsius.

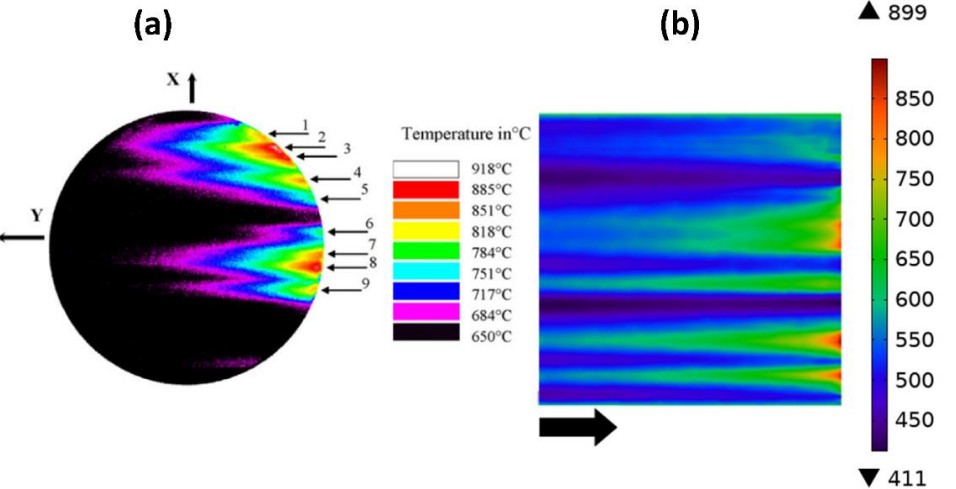

**Figure 10.** Comparison between the measured temperature field for the hole of test B obtained by [18] (license No.:5264161172347) shown in (**a**) and the hole of simulation B shown in (**b**). The surfaces are 1 and 2, and the simulated and test time is 650 μs. The temperature has units in degree Celsius.

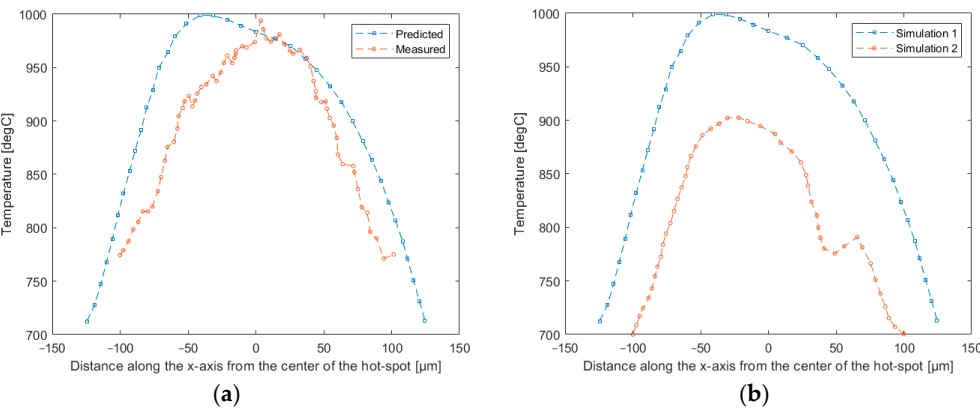

**Figure 11.** The comparison between the measured (test A) and predicted temperature (simulation A) hotspot along the edge of the hole of streak 7 shown in (**a**) and the corresponding comparison between simulation A and simulation B shown in (**b**). The surfaces used are 1 and 2 and the hotspot corresponds to the maximum flash temperature observed in the hole. The simulated and test time is 650 μs.

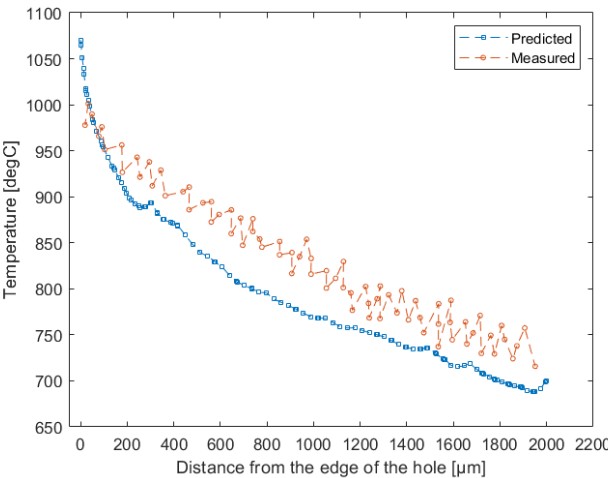

**Figure 12.** Temperature evolution along the sliding direction away from the hotspot of simulation A and test A for streak 7. The simulated time is 650 μs.

## 4. Conclusions

A multi-scale flash temperature model was applied and validated against existing experimental work. The model predictions show that it is possible to accurately predict the formation of hotspots and the maximum temperatures that occur within the interface of sliding systems. The main strength of using the presented model is that it only assesses the flash temperature on a small portion of the whole contact region. From the presented results, a few conclusions can be drawn:

- It is possible to accurately predict the flash temperature rise in sliding systems.
- For a given surface topography with certain roughness parameters, it is possible to accurately predict the shape and magnitude of the different hotspots.
- It is sufficient to only assess the flash temperature on a single cell and not the entire contact region. This is important because it saves both computational power and memory.

**Author Contributions:** Conceptualization, J.C. and R.L.; methodology, J.C.; software, J.C.; validation, J.C., A.A. and R.L.; formal analysis, J.C.; investigation, J.C.; resources, R.L.; data curation, J.C.; writing—original draft preparation, J.C.; writing—review and editing, R.L. and A.A.; visualization, J.C.; supervision, R.L. and A.A.; project administration, R.L.; funding acquisition, R.L. All authors have read and agreed to the published version of the manuscript.

**Funding:** This research was funded by the Swedish Research council grant number 2019-04293 and 2020-03635.

**Institutional Review Board Statement:** Not applicable.

**Informed Consent Statement:** Not applicable.

**Data Availability Statement:** Not applicable.

**Acknowledgments:** The authors would like to acknowledge the financial support from the Swedish Research Council (VR) grant number 2020-03635 and 2019-04293.

**Conflicts of Interest:** The authors declare that they have no conflict of interest.

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
