# Peer review of "Validation of a Multi-Scale Contact Temperature Model for Dry Sliding Rough Surfaces"

_lubricants, doi:10.3390/lubricants10030041_

Round 1
Reviewer 1 Report
This manuscript is about validation of a multi-scale flash temperature model for sliding systems. Such topic was taken up because there is a lack of validated models and the accurate prediction of flash temperatures can help reduce thermomechanical failures. The Introduction provides a sufficient background to the topic and is supported by up-to-date references. The known models and their limitations are presented there. The research is well planned. The authors used their model described in Tribol. Lett. to predict flash temperatures and compared them with temperatures measured by Sutter & Ranc. In the results and discussion section, the authors use the term simulation A (p. 8, line 225) and simulation 1 (p. 8, line 233) alternately. I would suggest unifying the notation because the term simulation 1 is not defined in the text of the manuscript. Additionally, the white dashed box, mentioned in the caption for Figure 4, is not marked in the figure.
Taking into account the language of the manuscript, in the methodology section, the present tense is used alternately with the past tense. Additionally, in some places the English language needs to be improved:
- p. 1, lines 9-10: it should be “…prediction of the flash temperature…” instead of “…prediction the flash temperature…”
- p. 4, line 129-130: it should be “…conditions were achieved…” instead of “…conditions was achieved…”
- p. 4, line 132: it should be “…a hole with a diameter of 2 mm…” instead of “…a hole with diameter 2 mm…”
- p. 4, line 153: it should be “…conditions could be reached…” instead of “…conditions reached…”
- p. 5, line 169: it should be “…was made…” instead of “…has made…”
- p. 6, line 170: it should be “…the cell’s micro-scale contact region…” instead of “…the cells micro-scale contact region…”
- p. 6, line 176: it should be “…according to the equation…” instead of “…according the equation…”
- p. 8, line 227: it should be “…the surface roughness is chosen.” instead of “…the surface roughness are chosen.”
- p. 9, line 249: it should be “The maximum of measured and predicted temperatures are…” instead of “The maximum measured and predicted temperatures are…”
- p. 9, line 254: it should be “…is observed…” instead of “…if observed…”
- p. 9, line 254: it should be “The maximum of measured and predicted temperatures are…” instead of “The maximum measured and predicted temperatures are…”
- p. 10, line 283: it should be “…for the hole of test A…” instead of “…for hole of test A…”
- p. 11, line 302: it should be “…the formation of hot-spots…” instead of “…the formation hot-spots…”
In general, the manuscript is well structured, the conclusions are supported by the research results but the English language needs to be improved. The manuscript can be accepted for publication in Lubricants after minor revision.
Author Response
Dear Reviewer, we would like to first and foremost thank you for your time and comments to our work. We appericate this very much and we will try our best to address them.
We have now gone through the language and made significant changes as suggested in the review.
Reviewer 2 Report
- As surface roughness is influential, it is suggested to consider the roughness originated from various machined methods such as milling, scraping, or grinding.
- Will the hole with a diameter of 2 mm placed in the middle of the upper body, which is used to capture the flash temperature, affect the simulation results as it changes the structure of the system?
- The author uses 400000 elements in simulation, while the author should report what is the number of used elements of similar articles. Also, what are the effects of element number on results?
- In Tab.1, the two conditions are too approximate that the results of Fig. 8(a) and (b) are almost the same.
- The author declares that the prediction results of the model are promising, nevertheless, the predicted values are always larger than measured ones in Fig. 11a and the predicted values are less in Fig. 12. What is the reason?
Author Response
Response to Reviewer 2 Comments
Dear Reviewer, we would like to first and foremost thank you for your time and comments to our work. We appericate this very much and we will try our best to address them.
1. As surface roughness is influential, it is suggested to consider the roughness originated from various machined methods such as milling, scraping, or grinding.
Response 1: We have now added a brief explanation in which we clarify that we have chosen surface topographies with Weibull height probablity distribution. This type of topography is commonly used for modelling surfaces which are grinded.
2. Will the hole with a diameter of 2 mm placed in the middle of the upper body, which is used to capture the flash temperature, affect the simulation results as it changes the structure of the system?
Response 2: Actually the hole must be present for correctly model the system. Because the hole is there in the experiments (from ref. 17) too so we have to re-create this in the simulations as well.
3.The author uses 400000 elements in simulation, while the author should report what is the number of used elements of similar articles. Also, what are the effects of element number on results?
Response 3: We have now added a explanation in which we want to confirm that the element number of 400 000 was adequate for this problem. This number of elements does not affect the results and the problem is actually not so mesh dependent from the beginning.
4. In Tab.1, the two conditions are too approximate that the results of Fig. 8(a) and (b) are almost the same.
Response 4: Yes this is correct. Actually these were directly taken from ref. 17 (experiments). They only provided thermal counturs for two different tests (test A & B as we refer them in our paper). So we had no choice but to work with these two tests.
5. The author declares that the prediction results of the model are promising, nevertheless, the predicted values are always larger than measured ones in Fig. 11a and the predicted values are less in Fig. 12. What is the reason?
Response 5: Actually we think that the reason for this is because we believe that it would be difficult to correctly measure the temperature “tail” shown in Figure 12. This is something that the authors from ref. 17 also briefly talk about. As we already explained in the discussions section, the thermal radiation waves emitted from the stationary body may interfere with the measurements, causing it to read higher values than what it is in reality. We think this is the main reason, so we adressed it already from before.
Round 2
Reviewer 2 Report
The paper can be accepted for publication.